# Do family physicians with focused practice or Care of the Elderly training practice differently than others? A population-based, propensity score-matched cohort study

Rebecca H. Correia[1,2]*, David Kirkwood[3], Aaron Jones[1,3], Henry Yu-Hin Siu[4], Meredith Vanstone[4], Steve Slade[5], Andrew P. Costa[1,6]

1 Department of Health Research Methods, Evidence and Impact, Faculty of Health Sciences, McMaster University, Hamilton, Ontario, Canada, 2 Department of Family Medicine, Faculty of Medicine, Dalhousie University, Halifax, Nova Scotia, Canada, 3 ICES, McMaster University, Hamilton, Ontario, Canada, 4 Department of Family Medicine, Faculty of Health Sciences, McMaster University, Hamilton, Ontario, Canada, 5 College of Family Physicians of Canada, Mississauga, Ontario, Canada, 6 Centre for Integrated Care, St. Joseph's Health System, Hamilton, Ontario, Canada

* correirh@mcmaster.ca

## Abstract

### Background

Family physicians play a key role in the care of older adults, yet the impact of additional geriatric training or focused practice remains unclear.

### Objective

We compared performance on established clinical practice measures among family physicians with/without evidence of elderly-focused practice or training.

### Methods

We used linked administrative data to conduct a population-based, propensity score-matched cohort study. Participants included family physicians in Ontario, Canada with rostered patients in 2019. Using logistic regression, we established propensity scores to match physicians with a focused alternative funding plan and/or a Certificate of Added Competence in 'Care of the Elderly' at a 1:4 ratio to a control group without focused practices or added competence certificates. We compared 11 practice-based measures endorsed by Canadian clinicians and researchers, adjusting for physician factors, medical practice characteristics, and primary care activities.

### Results

We identified 232 family physicians with elderly-focused practices or training and 928 comparable controls. While differences in study group clinical practices were not statistically significant for most processes, they were in three areas. More physicians

**Data availability statement:** The dataset from this study is held securely in coded form at ICES. While legal data sharing agreements between ICES and data providers (e.g., healthcare organizations and government) prohibit ICES from making the dataset publicly available, access may be granted to those who meet pre-specified criteria for confidential access, available at www.ices.on.ca/DAS (email: das@ices.on.ca). Please note that the computer programs may rely on coding templates or macros that are unique to ICES and are therefore either inaccessible or may require modification. As a Prescribed Entity under Ontario's Personal Health Information Protection Act (PHIPA) and the Coroners Act, ICES has the authority to collect and use personal health information for specific purposes. PHIPA Section 45 provides Prescribed Entities with the authority to collect and use data to assist the government in the planning and management of the health system, and PHIPA section 44 provides Prescribed Entities with the authority to disclose data to third-party researchers. Therefore, ICES is bound by contracts, data sharing agreements, and research ethics standards, limiting the full dataset creation plan and underlying analytic code to only be available by contacting www.ices.on.ca/DAS (email: das@ices.on.ca). More information on data privacy in general can be found on the ICES website (https://www.ices.on.ca/data-privacy/).

**Funding:** RHC was supported by a Canadian Institutes of Health Research Canada Graduate Scholarship (funding reference #181540). MV is supported by a Canada Research Chair (Tier 2) in Ethical Complexity in Primary Care and APC is supported by a Canada Research Chair (Tier 2) in Integrated Care for Seniors. The funders had no role in study design, data collection and analysis, decision to publish, or preparation of the manuscript.

**Competing interests:** The authors have declared that no competing interests exist.

with elderly-focused practice or training conducted testing aligned with the most recent Canadian Consensus on Dementia and were more likely to prescribe potentially inappropriate medications and antipsychotics to older attached patients.

## Conclusions

We observed limited to no differences in clinical practice measures between family physicians with 'Care of the Elderly' focused practice or certification to those without. The lack of differences may reflect true performance, the effect of uniform constraints of primary care practice, or inherent limitations of objective performance measurement.

## Introduction

Primary care of older adults is complex due to multimorbidity, chronicity, polypharmacy, and the need for care integration across settings and multiple providers. [1–3] In Canada, older adults constitute a large proportion of family physicians' (FPs) overall medical practice and, compared to specialists, FPs provide the majority of older adult care. [4–6] Older adults' use of primary care services is expected to increase given demographic shifts, [7] but FPs vary in their confidence and skillset to care for older patients. [8–10] Physicians report interpersonal challenges, administrative burdens, inadequate time and remuneration, and gaps in knowledge of community resources/services as barriers to caring for older adults [4,10,11].

Family medicine is faced with adapting to the changing needs of aging populations, but FPs often struggle to deliver comprehensive, continuous, and coordinated care to older patients. [12,13] While all FPs achieve foundational knowledge and clinical skills to care for older adults, [14] some pursue additional training to hone geriatric competencies [15,16] and/or dedicate a portion of their medical practice exclusively to older patients. [17] At this time, the impacts of elderly-focused practice and training are largely unknown, [18] although some descriptive work has characterized these providers. [19–28] Recent advances have allowed FPs with focused practice or additional training to be identified within administrative data, enabling new analyses of their practice. [26] However, it is unclear whether these physicians with focused practice or additional training have different practice patterns that demonstrate an advanced level of care to older patients.

We previously established consensus on measurable practice-based activities that are relevant to caring for older adults. [29] Endorsed by a panel of Canadian clinicians and researchers, we developed technical specifications for the established performance measures by referencing a population-based, health administrative data source. In this study, we aimed to compare the clinical practice of FPs with/without focused practice or additional training to care for older adults on the consensus-based practice measures. We hypothesized that elderly-focused practice and enhanced geriatric training would contribute to variations in clinical activities.

## Materials and Methods

### Design/setting

We conducted a population-based, propensity score-matched cohort study to compare FP clinical practice in Ontario, Canada. Our reporting concords with REporting of studies Conducted using Observational Routinely collected health Data (S1 Appendix in S1 File) [30].

### Data source

We accessed multiple datasets at ICES, an independent, non-profit research institute that collects and analyzes health care and demographic data about publicly-funded encounters (S2 Appendix in S1 File). [31] Datasets were linked using unique encoded identifiers and analyzed via the Remote Access Environment. We conducted a project-specific data linkage with the College of Family Physicians of Canada (CFPC) Membership Database to identify FPs with a 'Care of the Elderly' (COE) Certificate of Added Competence (CAC). The authors did not have access to information that could potentially identify individual participants.

### Participants

We established a cohort of FPs who submitted at least one Ontario Health Insurance Plan (OHIP) fee claim in the calendar year, 2019. We excluded non-residents of Ontario and physicians whose main specialty did not indicate family medicine. FPs without rostered patients were subsequently excluded, as the performance measures pertain to the ongoing care of attached patients [32].

We identified FPs with a focused practice billing designation and/or additional training to care for older adults based on a classification developed previously by our team. [26] A focused practice designation was attributed to FPs who bill OHIP fee codes eligible to those enrolled in an alternative funding plan by the Ministry of Health. To apply for a focused practice billing designation, FPs demonstrate need within their community, relevant training/qualifications, and evidence of dedicating >20% of their practice time to caring for older patients. [33] Additional training was demonstrated by holding a COE CAC from the CFPC. [34] COE CAC holders have achieved a defined level of competence in caring for older adults across 18 Priority Topics. [15,35] There may be some overlap between CAC holders and focused practice FPs, but the groups are not identical [26].

### Outcomes

The Donabedian model – the dominant health services paradigm to assess, evaluate, and improve quality – conceptualizes the interrelationships of structures and processes affecting outcomes. [36] Process measures include interactions between patients and health care providers for clinical practice activities or service provision. Per Donabedian's model, we can identify and quantify FP practice activities that serve as process factors. [37] While these factors are relevant to quality measurement, they do not by themselves constitute complete measures of quality.

In this study, we trialed and refined the measurement of 11 practice-based processes endorsed using the RAND/UCLA Appropriateness Method (RAM). [29] These established measures characterize appropriate and important primary care activities for older across four COE Priority Topics. [35] In the RAM study, health services researchers (RHC, AJ, APC) drafted technical definitions corresponding to endorsed measures. Panelists reviewed the specifications in a synchronous group meeting and rated items in two questionnaires. The technical definitions examined in this work reflect panelists' feedback and review by a physician (HS) for clinical accuracy.

Although 12 processes were endorsed in the RAM study, we assessed 11 outcomes here due to considerable overlap in the measurement and interpretation of two items. S3 Appendix in S1 File details the relevant dataset/variable names and steps in computation. For each process, the presence or absence of the practice activity was determined and

summarized as a proportion at the physician-level. This approach resulted in each measure representing the average proportion of FPs who delivered/performed the practice-based activity to attached older patients. Missing data resulted in the exclusion of matched individuals (S4 Appendix in S1 File).

## Analysis

We used propensity score matching to identify a comparable control group of FPs without evidence of a COE focused practice designation or CAC. These FPs may represent those with naturally aged practices [38] and/or physicians who acquired additional geriatric competencies through educational opportunities other than CAC training. [39–42] Propensity scores estimate the probability of assignment to an exposure and are used to balance baseline characteristics between two otherwise non-comparable groups. [43] Matching on propensity scores enabled us to reduce the effects of confounding on estimates of the association between our exposure (i.e., having a focused practice or CAC) and practice measures.

We calculated propensity scores using logistic regression to model the probability of exposure while adjusting for physician factors (i.e., years in practice, community size of primary practice location), medical practice characteristics (i.e., practice type and patient enrolment model), and primary care activities (i.e., long-term care [LTC] practice and number of patients cared for aged ≥65). These matching variables were selected based on their relevance to family medicine practice organization, care for older adults, and significant differences at baseline. FPs with evidence of elderly-focused practice or additional training were matched at a 1:4 ratio to controls without replacement using a caliper of 0.2 times the standard deviation of the logit of the propensity score [44–46].

We reported provider and practice characteristics using measures of general frequency, central tendency, dispersion, and standardized differences. [47] We reported mean differences to compare rates between groups. We then compared practice differences between FP groups by conducting t-tests with the conventional alpha = 0.05 threshold for significance. In a sensitivity analysis, we compared practice differences between FPs with evidence of elderly-focused practice versus additional training. All analyses were performed using Statistical Analysis Software, version 9.4. Data were accessed between February 22, 2024 and March 15, 2024 for research purposes.

## Ethics

This study was approved by the Hamilton Integrated Research Ethics Board (11391). The use of ICES data is authorized under Section 45 of Ontario's Personal Health Information Protection Act, which allows for the collection and analysis of health care and demographic data, without consent, for health system evaluation and improvement.

## Results

12,701 FPs met our selection criteria and 232 (1.83%) had evidence of an elderly-focused practice or CAC (Fig 1). Of the 232 FPs, 94 (40.52%) had COE CACs without a focused practice, 98 (42.24%) had evidence of a focused practice but no COE CAC, and 40 (17.24%) had both a COE CAC and focused practice. Propensity score-matching resulted in 232 FPs with an elderly-focused practice or CAC and 928 controls (n = 1,160).

Before matching, the groups were imbalanced on numerous covariates. After matching, the standardized differences for all descriptive factors were ≤0.1, suggesting comparability between groups (Table 1). FPs with COE focused practice designations or training tended to practice comprehensively (n = 146; 62.93%) with 382.57 (SD = 249.57) unique encounters, on average, with patients aged ≥65. Ninety-one exposed FPs (39.22%) were most responsible physicians in LTC.

We observed significant differences on three practice-based measures on the basis of elderly-focused practices and CACs (Table 2). On average, FPs with elderly-focused practices and CACs performed more testing for older patients aligned with the most current Canadian Consensus on Dementia (mean difference: 1.82%; CI = 0.44–3.20) compared to controls (Indicator 3). In contrast to other FPs, more COE physicians with focused practices and CACs prescribed one or more potentially inappropriate prescriptions (PIPs) to a greater proportion of attached older patients (mean difference:

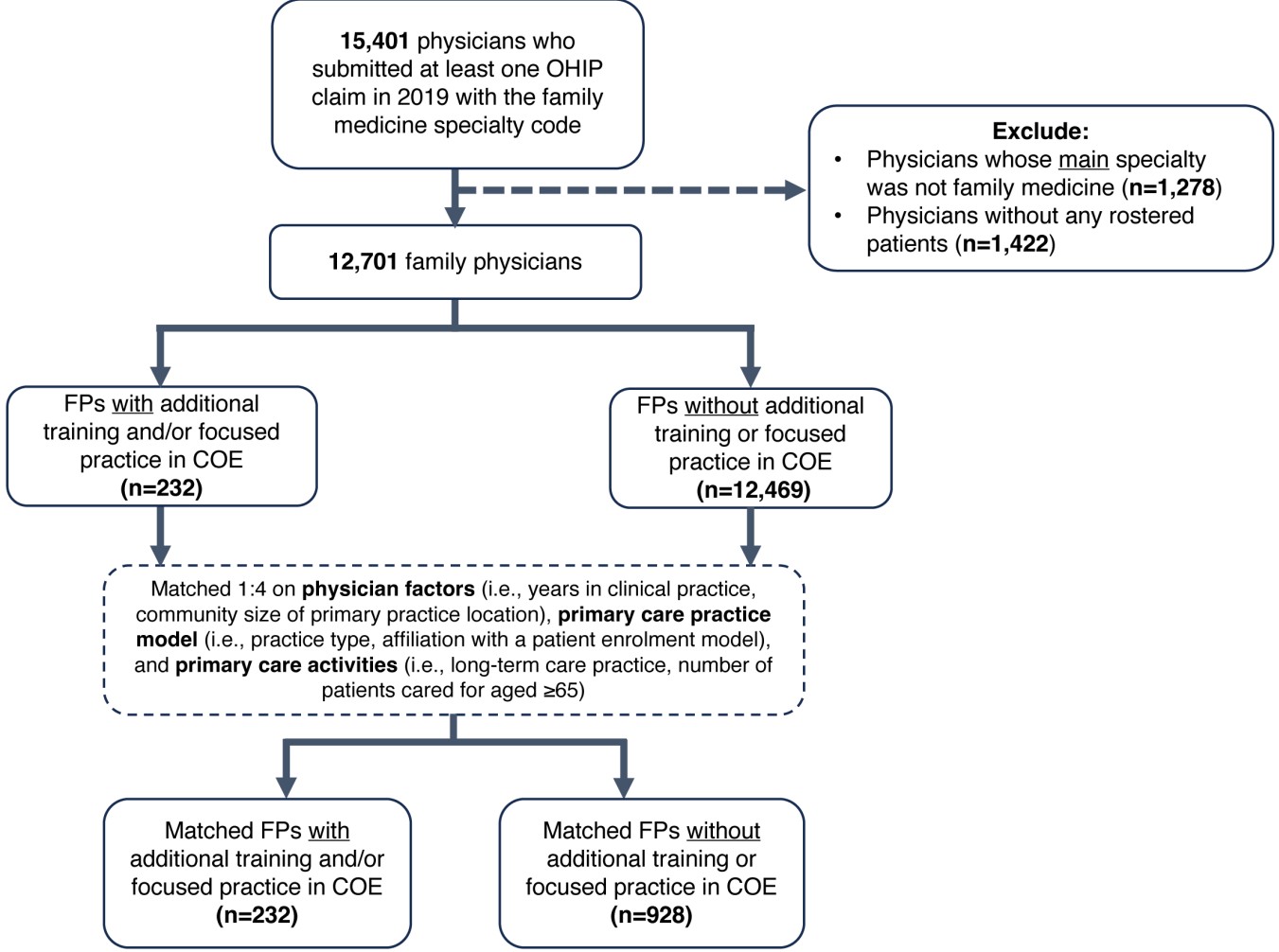

**Fig 1. Flow diagram of participant selection and matching.** OHIP = Ontario Health Insurance Plan; FP = family physician; COE = Care of the Elderly.

8.35%; CI = 4.94–11.76) (Indicator 7). Further, for attached older patients living with dementia, FPs with elderly-focused practices or training prescribed more antipsychotics (mean difference: 5.00%; CI = 1.73–8.28) versus controls (Indicator 10). We observed no significant differences for the remaining processes (n = 8).

In contrast to the main findings, we identified significant differences between the elderly-focused FP sub-groups on the basis of managing medical conditions (Indicators 1–4) and reporting potential driving issues (Indicator 11) (S5 Appendix in S1 File).

## Discussion

We observed similarities on eight of eleven clinical practice measures between FPs with focused practice or certification in COE to those without additional training or focused practice. The lack of differences for most performance measures may suggest that FPs who pursue focused practices or CACs in COE function like other FPs whose medical practice comprises a comparable number of older adults and levels of LTC activity. Similarities in clinical practice between the groups may reflect that additional performance is difficult to accomplish in the constraints of practice that limit comprehensive

**Table 1. Participant characteristics after matching (n = 1,160).**

| | FPs with a focused practice or certification in COE (n = 232) | FPs without a focused practice or certification in COE (n = 928) | Standardized difference |
|---|---|---|---|
| | N (%) a | | |
| **Physician Demographics** | | | |
| **Years in practice** a | 18.82 (11.29) | 19.01 (13.10) | 0.02 |
| **Community size** b | | | |
| ≥1,500,000 residents | 89 (38.36) | 358 (38.58) | 0.004 |
| 500,000–1,499,999 residents | 45 (19.40) | 189 (20.37) | 0.02 |
| 100,000–499,999 residents | 58 (25.00) | 198 (21.34) | 0.09 |
| 10,000–99,999 residents | 22 (9.48) | 107 (11.53) | 0.07 |
| <10,000 residents | 18 (7.76) | 76 (8.19) | 0.02 |
| **Primary Care Practice Model** | | | |
| **Full-time affiliation with a PEM** | 155 (66.81) | 628 (67.67) | 0.02 |
| **Medical practice type** | | | |
| Comprehensive | 146 (62.93) | 569 (61.31) | 0.03 |
| Focused | 53 (22.84) | 207 (22.31) | 0.01 |
| Other or <44 days in practice | 33 (14.22) | 152 (16.38) | 0.06 |
| **Primary Care Activities** | | | |
| **MRP in long-term care** | 91 (39.22) | 370 (39.87) | 0.01 |
| **Primary Care Patients** | | | |
| **Patients with a health services encounter** a | | | |
| Total aged ≥65 | 382.57 (249.57) | 386.88 (297.08) | 0.02 |

FP = family physician; COE = Care of the Elderly; PEM = patient enrolment model; MRP = most responsible physician.

a Mean and standard deviation are reported for continuous variables.

b Based on Census Metropolitan Area size.

care and follow-up. Another possibility is that FPs without focused practices or CACs who are in practice longer may attain elderly-related competencies over time, resulting in comparable skillsets. Clinical activities that did not demonstrate differences, such as immunization schedules or medication reviews, may reflect similar levels of expertise developed in foundational family medicine training to manage risks. Both FPs with focused practice or enhanced training and matched controls demonstrated some increased interest or commitment to care for older adults – signifying the important roles of all FPs in caring for older adults.

Slightly higher rates of consensus-based dementia testing and prescribing potentially inappropriate and antipsychotic medications were observed among FPs with elderly-focused practices or enhanced training. These processes – safe prescribing and adhering to guidelines – are emphasized in enhanced skills training and may reflect the expertise of FPs who self-designate a medical practice focused on aged patients. Seven Key Features for COE CAC assessment relate to appropriate prescribing, [35] suggesting greater knowledge/awareness of PIP risks. Our observed practice differences may be explained by FPs with elderly-focused practices or CACs having greater comfort with the use of PIPs and anti-psychotics, which may be warranted or beneficial in some circumstances. [48] Further, these physicians are more likely to care for patients with greater medical complexity (i.e., frailty, cognitive impairment, multimorbidity) and possess additional expertise to manage and understand the nuances of appropriately prescribing PIP to older patients. [49,50] Our study data could not control for patient-level factors (e.g., sex, multimorbidity) that have been shown to increase PIP use.

**Table 2. Clinical activities of family physicians with/without elderly-focused practices or certification.**

| | Mean (SD) | | Mean difference (95% CI) | P value |
|---|---|---|---|---|
| | FPs with a focused practice or certification in COE | FPs without a focused practice or certification in COE | | |
| **Medical Conditions** | | | | |
| **Indicator 1:** Proportion of attached patients aged ≥65 who received the influenza vaccine, % | 30.02 (25.93) | 30.53 (24.60) | −0.51 (−4.72, 3.70) | 0.8132 |
| **Indicator 2:** Proportion of attached patients aged ≥65 living with COPD who received influenza and pneumococcal immunizations, % | 31.94 (25.50) | 32.24 (25.81) | −0.30 (−5.33, 4.74) | 0.9073 |
| **Indicator 3^:** Proportion of attached patients aged ≥65 living with dementia who received tests aligned with the most current Canadian Consensus on Dementia, % | 4.42 (7.40) | 2.60 (6.52) | 1.82 (0.44, 3.20) | 0.0097* |
| **Indicator 4^:** Proportion of attached patients aged ≥65 living with dementia who received dementia care management, % | 70.21 (27.10) | 68.75 (27.58) | 1.46 (−4.19, 7.11) | 0.6120 |
| **Appropriate Prescribing** | | | | |
| **Indicator 5:** Proportion of attached patients aged ≥65 who are prescribed one or more benzodiazepines, % | 9.81 (10.55) | 8.95 (9.69) | 0.86 (−0.82, 2.53) | 0.3154 |
| **Indicator 6:** Proportion of attached patients aged ≥65 who are prescribed one or more medications with strong anticholinergic effects, % | 0.07 (0.26) | 0.11 (0.55) | −0.04 (−0.12, 0.05) | 0.3722 |
| **Indicator 7:** Proportion of attached patients aged ≥65 who are prescribed one or more potentially inappropriate medications (e.g., from Beers list, START/STOPP criteria), % | 38.80 (21.74) | 30.45 (19.73) | 8.35 (4.94, 11.76) | <.0001* |
| **Indicator 8:** Proportion of attached patients aged ≥65 with more than one prescribing physician who received a collaborative medication review, % | 0.06 (0.21) | 0.09 (0.51) | −0.03 (−0.11, 0.05) | 0.4631 |
| **Indicator 9:** Proportion of attached patients aged ≥65 living with CHF who were prescribed ACE inhibitors, ARBs, beta-blockers, or SGLT2 inhibitors, % | 59.09 (23.61) | 57.15 (25.22) | 1.95 (−3.34, 7.23) | 0.4702 |
| **Indicator 10^:** Proportion of attached patients aged ≥65 living with dementia who are prescribed antipsychotics, % | 20.18 (18.74) | 15.17 (15.15) | 5.00 (1.73, 8.28) | 0.0028* |
| **Driving Issues** | | | | |
| **Indicator 11^:** Proportion of attached patients aged ≥65 living with dementia whose medical condition was reported to the Ministry of Transportation, % | 1.68 (4.94) | 0.98 (2.99) | 0.70 (−0.01, 1.41) | 0.0542 |

FP = family physician; COE = Care of the Elderly; COPD = chronic obstructive pulmonary disease; CHF = congestive heart failure; ACE = Angiotensin-converting-enzyme; ARBs = Angiotensin receptor blockers; SGLT2 = Sodium-glucose cotransporter-2.

*Significant at the level of 0.05.

^ Indicator also relates to the Cognitive Impairment 'Care of the Elderly' Priority Topic.

[51–54] Therefore, it is not possible to comment on the appropriateness of using these medications or to make inferences about their impacts on patient outcomes. If focused practice or CAC training enables FPs to better discern when to prescribe or not, then there is increased potential for improved patient outcomes.

We also observed greater adherence to testing aligned with the Canadian Consensus on Dementia among FPs with elderly-focused practices or CACs. For those FPs who earned CACs, this may point to a general strength of enhanced skills training in terms of increased awareness and alignment of practice with current evidence-based guidelines. While dementia care knowledge is required for family medicine certification, [14] COE CAC holders are expected to exhibit additional competence related to the nuances of assessing, diagnosing, and managing cognitive impairment. [35] Their

enhanced training and greater exposure/experience caring for older patients with memory concerns in a focused setting may explain the observed difference. FPs with a focused practice or CAC in COE may feel more able and, therefore, continue to care for patients as the effects of aging become more complex and require specialized knowledge. Future work can examine the impacts of elderly-focused practice and CACs on specialist referral patterns for patients with varying indications of cognitive impairment.

FPs with focused practices and CACs in COE are well-positioned to care for the rapidly aging and increasingly complex population of older adults who rely on community-based primary care. [4,55] Despite this growing and urgent need, COE CAC holders acknowledge limited incentives to establish focused practices, [18,19,56] which may be a factor limiting the potential impact of their added training. CAC holders are not expected to restrict or narrow their scope of practice to their domain of expertise; instead, enhanced skills training is considered an opportunity for FPs to extend their comprehensive skills while maintaining competence across the broad family medicine scope. [34] Prior estimates indicate that less than one-third of COE CACs directly support specialized services for older adults, such as memory clinics, rehabilitation units, retirement homes, and LTC. [21,26,27,57] While some FPs greatly contribute to specialized geriatric services, others maintain family practice models caring for patients of all ages. [26] Those who continue to care for patients across all life stages may see the diminishing influence of their specialized training for geriatric care, resulting in similar performance over time due to reduced exposure to higher-level geriatric presentations.

Beyond CAC training and focused practice, other practice considerations related to how FPs organize or are supported in their practice could be shaping care delivery and, in turn, performance on the assessed indicators. For example, preventative care incentives or team-based care models may result in similarities from targeted incentives and shared clinical responsibilities, irrespective of additional training or focused practice. Resumption of practice as a generalist after earning a CAC, with marginal remuneration incentives encouraging increased care of older patients or opportunities to apply domain-specific skills, may explain similarities in practice performance observed here. Our findings suggest that health care system infrastructure and remuneration are key influencers of FPs' future practice style, and could be barriers for FPs to make full use of their CAC and training. These insights point to the need to better align COE training with real-world practice environments, strengthen remuneration structures that support geriatric-focused care, and enhance the integration of geriatric expertise within primary care teams. Education programs alone cannot supersede structural barriers. Ensuring the sustainability and longevity of COE CACs and focused practice will require coordinated reforms in training, compensation, and primary care organization.

### Limitations

While the performance measures were valid in construction, [29] our assessment and interpretation of clinical practice was limited by available information within the data source. Given measurement challenges, the absolute indicator rates are knowingly imperfect, though the relative comparisons are robust. The performance metrics relied largely on physician remuneration data and Ontario Drug Benefit claims, which may not have been sensitive to distinguish practice differences. For measures with low rates in both groups, it is unknown whether these processes are not frequently occurring or if they reflect poor measurement.

In addition, we could not adjust for unmeasured confounders that may have affected practice differences (e.g., clinical indication) or disparities in access. Although the process indicators used in this study reflect meaningful aspects of FP practice in caring for older adults, they do not fully capture the downstream impacts of care, such as health service utilization or changes in patient function. As a result, our findings cannot directly link observed changes in practice activities to patient outcomes, highlighting the need for future research that examines whether differences in FP processes relate to measurable improvements in clinical outcomes for older adults. Lastly, FPs had an average practice duration of 19 years; the training and expectations of CAC graduates today differ from those two decades ago. Assessing practice activities that are more closely tied to COE expected competencies may elucidate performance differences.

## Conclusions

We observed limited to no differences in clinical practice measures between FPs with focused practice and certification in COE compared to FPs without additional training or focused practice. The lack of differences may reflect true performance, the effects of uniform constraints of primary care practice, or inherent limitations of objective performance measurement. Future work can examine the impacts of practice differences on variations in quality of care as measured by patient outcomes.

## Supporting information

**S1 File. Supporting Information File.** Contains the completed RECORD checklist (S1 Appendix), description of relevant datasets (S2 Appendix), technical definitions of process measures (S3 Appendix), summary of missing data (S4 Appendix), and sensitivity analysis (S5 Appendix).
(DOCX)

## Acknowledgments

This study was supported by ICES, which is funded by an annual grant from the Ontario Ministry of Health (MOH) and the Ministry of Long-Term Care (MLTC). This document used data adapted from the Statistics Canada Postal Code$^{OM}$ Conversion File, which is based on data licensed from Canada Post Corporation, and/or data adapted from the Ontario Ministry of Health Postal Code Conversion File, which contains data copied under license from ©Canada Post Corporation and Statistics Canada. The opinions, results and conclusions reported in this paper are those of the authors and are independent from the funding sources. No endorsement by ICES, the MOH or MLTC is intended or should be inferred. Parts of this material are based on data and/or information compiled and provided by CIHI, Ontario Health (OH) and the Ontario Ministry of Health. The analyses, conclusions, opinions and statements expressed herein are solely those of the authors and do not reflect those of the funding or data sources; no endorsement is intended or should be inferred. Lastly, we acknowledge the College of Family Physicians of Canada for facilitating a data linkage that enabled our study and thank IQVIA Solutions Canada Inc. for use of their Drug Information File.

The dataset from this study is held securely in coded form at ICES. While legal data sharing agreements between ICES and data providers (e.g., healthcare organizations and government) prohibit ICES from making the dataset publicly available, access may be granted to those who meet pre-specified criteria for confidential access, available at www.ices.on.ca/DAS (email: das@ices.on.ca). The full dataset creation plan and underlying analytic code are available from the authors upon request, understanding that the computer programs may rely upon coding templates or macros that are unique to ICES and are therefore either inaccessible or may require modification.

## Author contributions

**Conceptualization:** Rebecca H. Correia, Andrew P. Costa.

**Data curation:** David Kirkwood.

**Formal analysis:** Rebecca H. Correia.

**Funding acquisition:** Rebecca H. Correia.

**Investigation:** Rebecca H. Correia, David Kirkwood, Aaron Jones, Henry Yu-Hin Siu, Meredith Vanstone, Andrew P. Costa.

**Methodology:** Rebecca H. Correia, Aaron Jones, Henry Yu-Hin Siu, Meredith Vanstone, Andrew P. Costa.

**Project administration:** Rebecca H. Correia.

**Resources:** Rebecca H. Correia.

**Software:** Rebecca H. Correia.

**Supervision:** Meredith Vanstone, Andrew P. Costa.

**Validation:** Andrew P. Costa.

**Visualization:** Rebecca H. Correia.

**Writing – original draft:** Rebecca H. Correia.

**Writing – review & editing:** David Kirkwood, Aaron Jones, Henry Yu-Hin Siu, Meredith Vanstone, Steve Slade, Andrew P. Costa.

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
