## [Decision Letter · Decision Letter 0]

27 Nov 2025

PONE-D-25-34776Do family physicians with focused practice or Care of the Elderly training practice differently than others? A population-based, propensity score-matched cohort studyPLOS ONE

Dear Dr. Correia,

Thank you for submitting your manuscript to PLOS ONE. After careful consideration, we feel that it has merit but does not fully meet PLOS ONE’s publication criteria as it currently stands. Therefore, we invite you to submit a revised version of the manuscript that addresses the points raised during the review process.

This paper contains interesting content and is highly regarded. However, the reviewers have suggested several points that require revision. Please read them carefully and address them appopriately.

We look forward to receiving your revised manuscript.

Kind regards,

Masaki Mogi

Academic Editor

PLOS ONE

2. In the online submission form, you indicated that [The dataset from this study is held securely in coded form at ICES. While legal data sharing agreements between ICES and data providers (e.g., healthcare organizations and government) prohibit ICES from making the dataset publicly available, access may be granted to those who meet pre-specified criteria for confidential access, available at www.ices.on.ca/DAS (email: das@ices.on.ca). The full dataset creation plan and underlying analytic code are available from the authors upon request, understanding that the computer programs may rely upon coding templates or macros that are unique to ICES and are therefore either inaccessible or may require modification.].

Additional Editor Comments (if provided):

Reviewers' comments:

Reviewer's Responses to Questions

**Comments to the Author**

1. Is the manuscript technically sound, and do the data support the conclusions?

Reviewer #1: Yes

2. Has the statistical analysis been performed appropriately and rigorously? 

Reviewer #1: Yes

3. Have the authors made all data underlying the findings in their manuscript fully available?

Reviewer #1: Yes

4. Is the manuscript presented in an intelligible fashion and written in standard English?

Reviewer #1: Yes

5. Review Comments to the Author

Reviewer #1: This population-based, propensity score–matched cohort study examined whether FPs in Ontario with a COE Certificate of Added Competence or an elderly-focused practice deliver different clinical care compared to those without such specialization. Among 232 COE/focused FPs and 928 matched controls, significant differences were observed in only three of eleven practice indicators: dementia-related testing and higher prescribing of potentially inappropriate medications and antipsychotics (higher in COE group). Overall, clinical practice patterns were largely similar, suggesting structural constraints or limited sensitivity of administrative measures to detect nuanced differences in geriatric competence

This is a well-designed and methodologically sound population-based cohort study that explores an important and timely question in primary care geriatrics. The analysis is robust, and the findings are clearly presented. However, the minimal observed differences between groups highlight the challenges of evaluating the impact of specialized geriatric training using administrative indicators alone. Clarifying the study’s primary aim, refining exposure definitions, and strengthening the interpretation toward educational and policy implications would further enhance the manuscript’s clarity and relevance. Overall, this study makes a valuable and policy-relevant contribution to the field of geriatric primary care.

1. The study addresses an important question regarding whether additional geriatric training or focused practice changes primary care behavior. However, the manuscript should more clearly define its primary aim — whether it evaluates clinical performance differences or educational/policy impact. Please state this explicitly in the Introduction and Discussion.

2. The grouping of physicians with COE certification and those with focused practice into a single exposure group may obscure distinct effects. These two categories represent different professional profiles. Provide subgroup or sensitivity analyses distinguishing COE-only, focused-only, and dual-trained physicians.

3. The key finding of limited practice differences requires deeper interpretation. It may suggest that system-level constraints limit the impact of additional training. Discuss how these results inform future design of the CAC/COE programs, remuneration models, and integration of geriatric expertise in primary care.

4. While process indicators are informative, they may not fully capture clinical quality. The discussion would benefit from linking findings to potential downstream outcomes (e.g., hospitalization, medication-related harm, patient function). Add a short paragraph connecting measured processes to meaningful patient outcomes.

6. PLOS authors have the option to publish the peer review history of their article (what does this mean?). If published, this will include your full peer review and any attached files.

Reviewer #1: No

---

## [Author Response · Author response to Decision Letter 1]

20 Jan 2026

Response to Reviewers

Do Family physicians with focused practice or care of the elderly training practice differently than others? A population-based, propensity score-matched cohort study

PONE-D-25-34776

Review Comments to the Author

This population-based, propensity score–matched cohort study examined whether FPs in Ontario with a COE Certificate of Added Competence or an elderly-focused practice deliver different clinical care compared to those without such specialization. Among 232 COE/focused FPs and 928 matched controls, significant differences were observed in only three of eleven practice indicators: dementia-related testing and higher prescribing of potentially inappropriate medications and antipsychotics (higher in COE group). Overall, clinical practice patterns were largely similar, suggesting structural constraints or limited sensitivity of administrative measures to detect nuanced differences in geriatric competence

This is a well-designed and methodologically sound population-based cohort study that explores an important and timely question in primary care geriatrics. The analysis is robust, and the findings are clearly presented. However, the minimal observed differences between groups highlight the challenges of evaluating the impact of specialized geriatric training using administrative indicators alone. Clarifying the study’s primary aim, refining exposure definitions, and strengthening the interpretation toward educational and policy implications would further enhance the manuscript’s clarity and relevance. Overall, this study makes a valuable and policy-relevant contribution to the field of geriatric primary care.

Thank you for this comprehensive summary of our study and for your encouraging comments to strengthen this work. In response to the four areas of improvement you indicated, we have revised our manuscript accordingly.

1. The study addresses an important question regarding whether additional geriatric training or focused practice changes primary care behavior. However, the manuscript should more clearly define its primary aim — whether it evaluates clinical performance differences or educational/policy impact. Please state this explicitly in the Introduction and Discussion.

We appreciate the opportunity to clarify our main study objective. This study examines clinical practice differences between family physicians (FPs) on the basis of additional geriatric training and/or focused practice compared to other FPs without this training or area of focus. Our findings have implications for both Care of the Elderly (COE) training and focused practice based on the clinical practice differences observed. Based on your feedback, we have clarified this aim in our Introduction and Discussion.

Lines 69-72: In this study, we aimed to compare the clinical practice of FPs with/without focused practice or additional training to care for older adults on the consensus-based practice measures. We hypothesized that elderly-focused practice and enhanced geriatric training would contribute to variations in clinical activities.

Lines 259-260: We observed limited to no differences in clinical practice measures between FPs with focused practice and certification in COE compared to FPs without additional training or focused practice.

2. The grouping of physicians with COE certification and those with focused practice into a single exposure group may obscure distinct effects. These two categories represent different professional profiles. Provide subgroup or sensitivity analyses distinguishing COE-only, focused-only, and dual-trained physicians.

Thank you for this suggestion. We originally chose not delineate findings across the three physician sub-groups (CAC-only, focused-only, both COE and focused) due to concerns about how these findings describe small groups of physicians (ranging from 27-72 FPs).

However, as a sensitivity analysis in response to your comment, we added comparisons of the performance measures across these three groups in S5 Appendix. Ultimately, we did not identify clear patterns across the three sub-groups. For example, the CAC-only and both COE and focused groups approximate each other for some indicators, whereas the focused-only relates to the mean estimate for the “no evidence of focused practice or certification” group. However, observations such as these do not apply to all indicators within particular categories (e.g., medical conditions, appropriate prescribing, driving issues, cognitive impairment). Therefore, in our Results, we only commented on how the difference in significance for four indicators changed across the sub-groups.

Lines 144-146: In a sensitivity analysis, we compared practice differences between FPs with evidence of elderly-focused practice versus additional training.

Lines 180-182: In contrast to the main findings, we identified significant differences between the elderly-focused FP sub-groups on the basis of managing medical conditions (Indicators 1 to 4) and reporting potential driving issues (Indicator 11) (S5 Appendix).

3. The key finding of limited practice differences requires deeper interpretation. It may suggest that system-level constraints limit the impact of additional training. Discuss how these results inform future design of the CAC/COE programs, remuneration models, and integration of geriatric expertise in primary care.

Thanks for your comment. We agree that the nominal differences observed between FP groups may reflect that despite having additional training or a greater focus/capacity to care for older adults, these FPs are limited by system-level factors. As said:

Lines 183-187: The lack of differences for most performance measures may suggest that FPs who pursue focused practices or CACs in COE function like other FPs whose medical practice comprises a comparable number of older adults and levels of LTC activity. Similarities in clinical practice between the groups may reflect that additional performance is difficult to accomplish in the constraints of practice that limit comprehensive care and follow-up.

We have since expanded on these implications in our interpretation:

Lines 227-228: Despite this growing and urgent need, COE CAC holders acknowledge limited incentives to establish focused practices,18,19,56 which may be a factor limiting the potential impact of their added training.

Lines 238-242: Beyond CAC training and focused practice, other practice considerations related to how FPs organize or are supported in their practice could be shaping care delivery and, in turn, performance on the assessed indicators. For example, preventative care incentives or team-based care models may result in similarities from targeted incentives and shared clinical responsibilities, irrespective of additional training or focused practice.

Lines 252-257: These insights point to the need to better align COE training with real-world practice environments, strengthen remuneration structures that support geriatric-focused care, and enhance the integration of geriatric expertise within primary care teams. Education programs alone cannot supersede structural barriers. Ensuring the sustainability and longevity of COE CACs and focused practice will require coordinated reforms in training, compensation, and primary care organization.

4. While process indicators are informative, they may not fully capture clinical quality. The discussion would benefit from linking findings to potential downstream outcomes (e.g., hospitalization, medication-related harm, patient function). Add a short paragraph connecting measured processes to meaningful patient outcomes.

The process indicators examined in this work were established in a consensus study conducted previously by our team. That study used the Donabedian Structure-Process-Outcome model to conceptualize the interrelationships of broader factors, physician activities, and impacts for older adult patients. We discuss our focus on processes in the following excerpt:

Lines 102-108: The Donabedian model – the dominant health services paradigm to assess, evaluate, and improve quality – conceptualizes the interrelationships of structures and processes affecting outcomes.36 Process measures include interactions between patients and health care providers for clinical practice activities or service provision. Per Donabedian’s model, we can identify and quantify FP practice activities that serve as process factors.37 While these factors are relevant to quality measurement, they do not by themselves constitute complete measures of quality.

We focused on processes because they directly relate to FP activities, whereas structures and outcomes are broader and reflect multiple factors (e.g., policies, resources, contributions of other health professionals). Therefore, we cannot connect the measured processes to meaningful outcomes, but we have added a reflection on this in our Limitations:

Lines 267-278: Although the process indicators used in this study reflect meaningful aspects of FP practice in caring for older adults, they do not fully capture the downstream impacts of care, such as health service utilization or changes in patient function. As a result, our findings cannot directly link observed changes in practice activities to patient outcomes, highlighting the need for future research that examines whether differences in FP processes relate to measurable improvements in clinical outcomes for older adults.

Also, as stated in our Conclusion, examining outcomes could be a future study building on this work:

Lines 280-281: Future work can examine the impacts of practice differences on variations in quality of care as measured by patient outcomes.

---

## [Decision Letter · Decision Letter 1]

10 May 2026

Do family physicians with focused practice or Care of the Elderly training practice differently than others? A population-based, propensity score-matched cohort study

PONE-D-25-34776R1

Dear Dr. Correia,

We’re pleased to inform you that your manuscript has been judged scientifically suitable for publication and will be formally accepted for publication once it meets all outstanding technical requirements.

Kind regards,

Marianne Clemence

Staff Editor

PLOS One

Additional Editor Comments (optional):

Reviewers' comments:

Reviewer's Responses to Questions

**Comments to the Author**

1. If the authors have adequately addressed your comments raised in a previous round of review and you feel that this manuscript is now acceptable for publication, you may indicate that here to bypass the “Comments to the Author” section, enter your conflict of interest statement in the “Confidential to Editor” section, and submit your "Accept" recommendation.

Reviewer #1: All comments have been addressed

2. Is the manuscript technically sound, and do the data support the conclusions?

Reviewer #1: Yes

3. Has the statistical analysis been performed appropriately and rigorously? 

Reviewer #1: Yes

4. Have the authors made all data underlying the findings in their manuscript fully available?

Reviewer #1: Yes

5. Is the manuscript presented in an intelligible fashion and written in standard English?

Reviewer #1: Yes

6. Review Comments to the Author

Reviewer #1: The authors well responded to the reviewer's comments. Current manuscript meets the quality for the publication in PLOS One.

7. PLOS authors have the option to publish the peer review history of their article (what does this mean?). If published, this will include your full peer review and any attached files.

Reviewer #1: No

---

## [Editor Report · Acceptance letter]

PONE-D-25-34776R1

PLOS One

Dear Dr. Correia,

I'm pleased to inform you that your manuscript has been deemed suitable for publication in PLOS One. Congratulations! Your manuscript is now being handed over to our production team.

Kind regards,

on behalf of

Dr Marianne Clemence

Staff Editor

PLOS One